# Propionate Converting Anaerobic Microbial Communities Enriched from Distinct Biogeochemical Zones of Aarhus Bay, Denmark under Sulfidogenic and Methanogenic Conditions

**DOI:** 10.3390/microorganisms8030394

**Published:** 2020-03-11

**Authors:** Derya Ozuolmez, Alfons J. M. Stams, Caroline M. Plugge

**Affiliations:** Laboratory of Microbiology, Wageningen University & Research, Stippeneng 4, 6708 WE Wageningen, The Netherlands; adaderya@gmail.com (D.O.); fons.stams@wur.nl (A.J.M.S.)

**Keywords:** propionate, marine sediment, syntrophy, sulfate-reducing bacteria, methanogenic Archaea, Aarhus Bay

## Abstract

The relationship between predominant physiological types of prokaryotes in marine sediments and propionate degradation through sulfate reduction, fermentation, and methanogenesis was studied in marine sediments. Propionate conversion was assessed in slurries containing sediment from three different biogeochemical zones of Aarhus Bay, Denmark. Sediment slurries were amended with 0, 3, or 20 mM sulfate and incubated at 25 °C and 10 °C for 514–571 days. Methanogenesis in the sulfate zone and sulfate reduction in the methane zone slurries was observed. Both processes occurred simultaneously in enrichments originating from samples along the whole sediment. Bacterial community analysis revealed the dominance of *Desulfobacteraceae* and *Desulfobulbaceae* members in sulfate-amended slurries incubated at 25°C and 10°C. *Cryptanaerobacter* belonging to the *Peptococcaceae* family dominated sulfate-free methanogenic slurries at 25°C, whereas bacteria related to *Desulfobacteraceae* were dominant at 10°C. Archaeal community analysis revealed the prevalence of different genera belonging to *Methanomicrobiales* in slurries incubated at different temperatures and amended with different sulfate concentrations. *Methanosarcinaceae* were only detected in the absence of sulfate. In summary, Aarhus Bay sediment zones contain sulfate reducers, syntrophs, and methanogens interacting with each other in the conversion of propionate. Our results indicate that in Aarhus Bay sediments, *Cryptanaerobacter* degraded propionate in syntrophic association with methanogens.

## 1. Introduction

The mineralization of organic matter in anoxic marine sediments is a sequential process, in which several intermediates are produced by fermentative bacteria. These intermediates, including acetate, propionate, and butyrate, are eventually degraded to carbon dioxide and methane. Sulfate reduction and methanogenesis are important terminal electron-accepting processes and these two processes are controlled mainly by the availability of sulfate [1]. In marine sediments, sulfate reduction is considered to be dominant over methanogenesis [2]. Propionate is one of the principal intermediates in anaerobic degradation and in the presence of sulfate it can be converted by a variety of marine sulfate-reducing bacteria (SRB) either completely to carbon dioxide or incompletely to acetate (Appendix A). These SRB belong to the families *Desulfobacteraceae*, *Desulfobulbaceae*, *Syntrophobacteraceae,* and *Peptococcaceae* [3,4,5]. *Desulfobulbus* is an incomplete oxidizer able to use propionate that is converted to acetate and carbon dioxide. The acetate can subsequently be oxidized by sulfate reducers such as *Desulfobacter* spp. In the absence of sulfate, propionate conversion to CH_4_ and CO_2_ takes place, and this process is only possible when syntrophic bacteria and methanogens cooperate [6] (Appendix A). Syntrophic fatty acid metabolism is one of the rate-limiting steps in organic carbon degradation but contributes significantly to the carbon flux in methanogenic environments [6]. Cappenberg demonstrated that sulfate reduction and methanogenesis were spatially separated in a fresh water column, and Reeburgh and Heggie reported that in marine sediments methane production occurred only at depths where sulfate reduction was limited by a depleted supply [7,8]. However, more recent studies demonstrated that both sulfate reduction and methanogeniesis co-occur in anoxic marine environments, especially where organic carbon concentrations are high [9,10]. The relative distribution of SRB, including sulfate reducers that can also grow as syntrophs (e.g., *Syntrophobacter* sp.), in Black Sea and Aarhus Bay sediments is similar in both the methane zone and the sulfate zone [4,5]. Where sediments are high in organic matter, sulfate is depleted at shallow sediment depths, and methane production will occur [11]. In the absence of sulfate, many SRB fermented organic acids and alcohols, producing hydrogen, acetate, and carbon dioxide, rely on hydrogen- and acetate-scavenging methanogens to convert organic compounds to methane [12].

Kendall et al. [13] described marine propionate- and butyrate-degrading syntrophs and suggested that syntrophic associations play an important role in the formation of methane reserves in marine sediments. Other studies have shown that microbial populations in shallow methanogenic sediments include members of *Syntrophobacteraceae* [14,15]. Krylova and Conrad [16] speculated that in rice paddies propionate was degraded within methanogenic bacterial aggregates both in the presence and the absence of sulfate and that propionate degraders operated either as sulfate reducers or as H_2_-producing syntrophs.

In this study, we investigated propionate conversion and microbial diversity involved in the sulfate, sulfate–methane transition, and methane zones of Aarhus Bay, Denmark. We set up batch slurries with and without sulfate addition and made long term incubations at 10 °C and 25 °C. In this way, we were able to assess the effect of sulfate concentration, sediment depth, and temperature on the bacterial and archaeal community that was enriched.

## 2. Materials and Methods

### 2.1. Sediment Sampling

Sediment was collected during a research cruise in May 2011, in Aarhus Bay, Denmark. The studied site, Station M1, is located in the central part of the Bay, at position 56°07′066′′ N, 10°20′793′′ E. The temperature in situ was ~9 °C and the water depth was 15 m. Two 3 m long gravity cores were retrieved; one of them was sectioned in 10 cm depth intervals for physical, chemical, and molecular analyses and the other one was kept intact in the core liners, in sealed gas-tight plastic bags containing AnaeroGen sackets (Oxoid) at 4 °C until further processed.

### 2.2. Sediment Pore Water Analysis

Methane, sulfate, and sulfide analyses from sediment pore water were performed on the sampling day at the laboratories of Center for Geomicrobiology, Aarhus University. The pore water concentrations of sulfate and methane were determined as described in [17]. Hydrogen sulfide was quantified in zinc-preserved porewater samples by the methylene blue method [18].

### 2.3. Sediment Slurry Incubations

Sediments from three different biogeochemical zones were used to establish replicate sediment slurries. Zones were defined based on sulfate and methane concentrations determined using pore water extracted from sediment during the sampling cruise. The sulfate concentration decreased from 18.5 mM at 15 cm of the core to a low background value of 0.1 mM at 170 cm. Methane increased steeply with depth below 120 cm and reached a plateau of 2 mM at 225 cm. The sediment core was divided into three pieces representing the sulfate zone (SR) (15–120 cm), the sulfate–methane transition zone (SMTZ) (120–170 cm) and the methane zone (MZ) (170–300 cm) (Appendix A). Stored sediment cores were processed under aseptic and anaerobic conditions in the laboratory. Subsamples representing a particular biogeochemical zone were mixed in an anaerobic chamber and used as inoculum for sediment slurry enrichments. An amount of 100 mL of the homogenized sediment from each zone was mixed with 300 mL of anaerobic mineral salts medium in 1 L serum bottles. The medium composition was (g/L): KH_2_PO_4_ (0.41), Na_2_HPO_4_ ·2H_2_O (0.53), NH_4_Cl (0.3), CaCl_2_.·2H_2_O (0.11), MgCl_2_·6H_2_O (3), NaHCO_3_ (4), Na_2_S·9H_2_O (0.024), KCL (0.5), NaCl (25). The medium was supplemented with 1 mL/L of acid trace element solution (50 mM HCl, 1 mM H_3_BO_3_, 0.5 mM MnCl_2_, 7.5 mM FeCl_2_, 0.5 mM CoCl_2_, 0.1 mM NiCl_2_, 0.5 mM ZnCl_2_), 1 mL/L of alkaline trace element solution (10 mM NaOH, 0.1 mM Na_2_SeO_3_, 0.1 mM Na_2_WO_4_, 0.1 mM Na_2_MoO_4_), and 10 mL/L vitamin solution (Biotin 20 mg/L, Nicotinamid 200 mg/L, p-Aminobenzoic acid 100 mg/L, Thiamin 200 mg/L, Pantothenic acid 100 mg/L, Pyridoxamine 500 mg/L, Cyanocobalamine 100 mg/L, Riboflavin 100 mg/L). Bottles were closed with butyl rubber stoppers and the headspace was exchanged with N_2_/CO_2_ (80:20%, *v/v*) and the final pressure was adjusted to 1.5 kPa. An amount of 10 mM propionate was used as a carbon source with and without 20 mM sulfate in sulfate zone and methane zone slurries, and with 3 mM and 20 mM sulfate for sulfate–methane transition zone slurries. The reason why we amended SMTZ slurries with 3 mM sulfate instead of 0 mM sulfate was to maintain the low sulfate concentration that is characteristic of SMTZ of Aarhus Bay. Control bottles were prepared in the same manner, without addition of propionate. One set of the bottles representing each condition in duplicate was incubated at 10 °C to mimic in situ temperature [19] and the other set was kept at 25 °C statically throughout the experiment. Regular liquid and gas sampling was performed to monitor substrate consumption, product formation, and to carry out molecular analysis. Regular additions of propionate and/or sulfate were included as soon as they were depleted to maintain the slurry conditions.

### 2.4. Analytical Methods

CH_4_ in the headspace of slurries was analyzed by gas chromatography as described previously [20]. Volatile fatty acids from centrifuged (10,000× *g*, 10 min) samples of the sediment slurries were analyzed by HPLC as described previously [20]. Data analyses were performed using ChromQuest (Thermo Scientific, Waltham, MA, USA) and Chromeleon software (Thermo Scientific, Waltham, MA, USA). Sulfate concentrations were analyzed by Ion Chromatography system equipped with an AS22 column (4 × 250 mm) and ED 40 electrochemical detector (Dionex, Sunnyvale, CA, USA). The eluents were 1.7 mM NaHCO_3_ and 1.8 mM Na_2_CO_3_. The analyses were conducted with a flow rate of 1.2 mL min^−1^ at 35 °C. Sodium bromide was used as internal standard. Sulfide measurements were done using the methylene blue method [18].

### 2.5. DNA Extraction

Genomic DNA was extracted from the sediment and enrichment slurry samples that were taken at different time points using the FastDNA SPIN Kit for Soil (MP Biomedicals, OH) according to manufacturer’s protocol. Adaptation of the commercial protocol was carried out to increase the DNA yield. An amount of 5 ml sediment or slurry sample was suspended in 10 mL of phosphate-buffered saline (PBS), sonicated at low power to detach cells from the solid phase and centrifuged at 4700× *g* for 20 min. The supernatant was discarded and the remaining pellet was re-suspended in 10 mL 0.5 M EDTA, pH 8, and incubated overnight at 4 °C to dissolve humic substances. After incubation, the suspension was centrifuged at 4700× *g* for 10 min, washed with PBS and DNA extraction procedures were applied to the pellet. The DNA was quantified with a Nanodrop ND-1000 spectrophotometer (Nanodrop Technologies, Wilmington, DE).

### 2.6. Bacterial 16S rRNA Gene Amplicon Pyrosequencing

Bacterial 16S rRNA gene fragments were amplified using barcoded primers covering the V1–V2 region of the bacterial 16S rRNA gene. The forward primer consisted of the 27F-DegS primer (5′- GTTYGATYMTGGCTCAG- 3′) appended with the titanium sequencing adaptor A (5′- CCATCTCATCCCTGCGTGTCTCCGACTCAG- 3′) and an 8-nucleotide sample specific error-correcting barcode at the 5′ end. An equimolar mix of two reverse primers was used, i.e., 338RI (5′- GCWGCCTCCCGTAGGAGT- 3′) and 338RII (5′- GCWGCCACCCGTAGG TGT- 3′) which carried the titanium adaptor B (5′- CCTATCCCCTGTGTGCCTTGGCAG TCTCAG- 3′) at the 5′ end. Sequences of both titanium adaptors were purchased from GATC Biotech (Konstanz, Germany). Genomic DNA was diluted to a concentration of 20 ng/µl based on Qubit® 2.0 fluorometer readings and amplicons were generated as described in [21]. Amplicons were sequenced using a 454 FLX genome sequencer in combination with titanium chemistry (GATC Biotech AG, Konstanz, Germany).

### 2.7. Analysis and Interpretation of the Pyrosequencing Data

Pyrosequencing data were analyzed using the Quantitative Insights Into Microbial Ecology (QIIME) 1.8.0 pipeline [22]. Sequence reads were initially filtered using default parameters and denoised [23] for removing low quality reads. UCHIME was used to remove chimeric sequences from pre-processed data from the dataset [24]. From the remaining set of high quality 16S rRNA gene sequences, operational taxonomic units (OTUs) were defined at a 97% identity level. A representative sequence from each OTU was aligned using PyNAST [25]. The taxonomic affiliation of each OTU was determined at an identity threshold of 97% using the UCLUST algorithm [24] and SILVA 111 database as a reference [26]. The relative amount of reads of every OTU to the total amount of reads per sample was quantified and the average relative amount of reads per representative OTU of each slurry sample was calculated. The demultiplexed reads of the 16S rRNA gene amplicon sequences were deposited at the European Nucleotide Archive (ENA) under study PRJEB36640 (*Bacteria*) and PRJEB36882 (*Archaea*).

### 2.8. Illumina HiSeq Analysis of Archaeal Community

Extracted DNA from the samples taken on the last incubation day from all slurries was used for archaeal community analysis. Barcoded amplicons were generated using a two-step PCR method that was shown to reduce the impact of barcoded primers on the outcome of microbial profiling as described in [27]. The 16S rRNA gene sequencing data were analyzed using NG-Tax, an in-house pipeline [28]. Paired-end libraries were filtered to contain only read pairs with perfectly matching barcodes, and those barcodes were used for demultiplex reads by the sample. Finally, operational taxonomic units were defined using an open reference approach, and taxonomy was assigned to those OTUs using the SILVA 16S rRNA gene reference database [29]. Microbial composition plots were generated using a workflow based on Quantitative Insights Into Microbial Ecology v1.2 [22]. The demultiplexed reads of the 16S rRNA gene amplicon sequences were deposited at the European Nucleotide Archive (ENA) under study PRJEB36640 (*Bacteria*) and PRJEB36882 (*Archaea*).

### 2.9. Statistical Analysis

Redundancy analysis was performed as implemented in the CANOCO 5 software package (Biometris, Wageningen, The Netherlands) in order to assess the extent of which experimental variables influenced the microbial community composition. The experimental variables tested were the incubation temperature, total concentrations of sulfate, propionate, acetate, and methane consumed/produced by the end of the incubations. A Monte Carlo permutation test based on 499 random permutations was used to determine which of the experimental variables significantly contributed to the observed variance in the composition of microbial communities at the order (for Bacteria) and family level (for *Archaea*). Orders and families of at least 5% relative abundance in any sample were included in the analysis. The community structure was visualized via ordination triplots with scaling focused on intersample differences. Correlations between bacterial and archaeal groups and experimental parameters were determined by means of the two-tailed Spearman’s Rank Order Correlation test using the statistical software SPSS Statistics (IBM SPSS Statistics, Version 22, IBM Corp., Armonk, NY, USA). A statistical significance level of 5% was applied.

## 3. Results and Discussion

### 3.1. Propionate Conversion in Sulfate Zone Sediment Slurries

Sulfate zone sediment slurries were incubated for 514 days at 25 °C and 10 °C. Total amounts of propionate and sulfate consumed, and acetate, sulfide, and methane produced in the slurries are listed in Appendix A. Propionate conversion in sulfate-amended slurries started immediately at 25 °C (Figure 1A). In contrast, it took 50 days in 10 °C slurries and the conversion was much slower along the incubation period (Figure 1B). Acetate and sulfide steadily increased as a result of repeated additions of propionate and sulfate, while acetate formation leveled off after 200 days at 25 °C (Figure 1A), but not at 10°C. Methane formation in sulfate-amended slurries was observed after 309 days at 25 °C (Figure 1A), while no methane was detected in slurries incubated at 10 °C (Figure 1B) throughout the study. At 25 °C between the days 220 and 430, propionate was converted coupled to sulfate reduction while the acetate formed was converted by acetoclastic methanogens, as no net acetate increase was observed (Figure 1A). The simultaneous increase in acetate and methane between the days 457 and 514 suggests syntrophic conversion of propionate by collaborative action of syntrophic bacteria and hydrogenotrophic methanogenic *Archaea*, in addition to the ongoing sulfate-dependent propionate conversion (Figure 1A).

In sulfate-free sediment slurries, several feeds of propionate led to the formation of methane and acetate until day 238 at 25 °C (Figure 1C). The fast propionate conversion with concomitant acetate and methane production at 25 °C suggests the activity of syntrophic propionate-converting consortia (Figure 1C). As a result of rapid propionate conversion, acetate accumulated in the slurries and was utilized only after propionate became depleted (Figure 1C). Similar conversion dynamics were reported for both methanogenic [30,31] and sulfidogenic batch cultures [32]. Apparently, acetoclastic methanogens could contribute to the conversion process only after propionate concentration became low in slurries. At 10°C a similar trend was observed but was much slower (Figure 1D).

### 3.2. Propionate Conversion in Sulfate–Methane Transition Zone Sediment Slurries

SMTZ sediments were treated with low (3 mM) and high (20 mM) concentrations of sulfate for 571 days at 25 °C and 10 °C. An amount of 3 mM sulfate concentration is characteristic for the SMTZ, therefore we aimed to maintain this concentration along the incubation in order to enrich for the zone-specific consortia. Propionate conversion coupled with sulfate reduction and consequently the sulfide and acetate concentrations increased in slurries treated with 20 mM sulfate (Figure 2A,B). Although the total amounts of consumed propionate and sulfate in high-sulfate slurries incubated at 10 °C and 25 °C was very similar (Appendix A), the products were different. Notably, concentrations of acetate accumulated in 25 °C slurries (9246 µmol/slurry) was much less than in 10 °C slurries (27,709 µmol/slurry) (Figure 2A,B, Appendix A), indicating the development of acetate-consuming consortia at higher incubation temperatures. Methane was observed earlier and reached a much higher concentration in 25 °C slurries compared to 10 °C slurries. Evaluating the measured substrates and products in high sulfate-amended SMTZ slurries, more sulfate was reduced than expected for incomplete propionate oxidation (Appendix A).

On the other hand, the consumed propionate and accumulated acetate concentrations in the slurries incubated with low sulfate concentrations at 25 °C (Figure 2C) and at 10 °C (Figure 2D) differed significantly. Methane formation started earlier and methane concentration was five times higher at 25 °C than at 10 °C. Evaluating the propionate degradation stoichiometry at 25 °C suggested that sulfate reducers, syntrophs, and methanogens were participating in the degradation (Appendix A, Figure 2C). Obviously, the low levels of sulfate were not sufficient to support complete propionate conversion to proceed solely through sulfate reduction and, consequently, syntrophic propionate-degrading communities became active.

### 3.3. Propionate Conversion in Methane Zone Sediment Slurries

The propionate conversions in sulfate-amended MZ and SZ slurries at 25 °C were similar in terms of immediate start of conversions (Figure 3A). The highest propionate concentration that was converted in the MZ slurries was measured under sulfate-amended conditions at 25 °C (Figure 3A). Assessing the stoichiometry of propionate conversion in MZ slurries pointed to incomplete propionate oxidation by sulfate reducers (Appendix A, reaction 2). Methane concentration rapidly increased after 414 days of incubation in MZ slurry (Figure 3A). The late methane production might be linked to sulfide inhibition of methanogens which was reported in previous studies [33]. Despite the presence of high amounts of sulfide, methanogenic *Archaea* adapted and were involved in the consumption of conversion products. On the other hand, the conversion of propionate at 10 °C was slow and no methane production occurred during the experiment (Figure 3B).

Propionate conversion dynamics in sulfate-free MZ slurries at 25 °C was similar to sulfate-free SZ slurries at 25 °C (Figure 3C). After 107 days, propionate conversion rapidly started with concomitant acetate and methane production. Acetate that accumulated until day 259 was almost completely consumed by the end of the experiment. The highest methane concentration among methane zone slurries was observed in the sulfate-free slurry at 25 °C (Figure 3C, Appendix A). Our observations validate the existence of efficient propionate conversion by tightly coupled syntrophic propionate converting, hydrogenotrophic, and acetotrophic methanogenesis reactions in the absence of sulfate at 25 °C. Propionate conversion in sulfate-free slurries at 10 °C occurred after, a lag phase of 226 days was observed, and acetate and methane was produced. It is known that microorganisms in subsurface environments have low metabolic activity and growth rate but can persist in a dormant state which enables them to survive under unfavorable conditions such as low substrate availability [34].

### 3.4. Bacterial Community Composition Revealed by Pyrosequencing Analysis

PCR amplified partial 16S rRNA gene fragments obtained from the original sediment samples belonging to each biogeochemical zone and the last sampling time of all slurries were sequenced. After filtering and trimming, between 1888 and 27196 high quality reads were found per sample (Appendix A) and clustered into 64–131 operational taxonomic units per sample. OTUs classified into 33 phyla, with 95% of the OTUs belonging to *Proteobacteria* (61.3%), *Firmicutes* (17.5%), *Chloroflexi* (9.8%), *Bacteroidetes* (3.4%), *Spirochaetes* (2.5%), and Candidate division OP9 (1%). The *Desulfobacteraceae* and *Desulfobulbaceae* were the two dominant families in the phylum *Proteobacteria*, having 38.6% and 29.2% of the reads, respectively (Appendix A). The *Firmicutes* phylum was dominated by the *Peptococcaceae* family comprising 86.8% of the reads associated with this phylum. Similarly, the *Anaerolinaceae* family dominated the *Chloroflexi* phylum representing 78.1% of the reads (Appendix A). One of the above mentioned four families were most dominant in each sample. Despite the overall dominance of these families, the relative abundances of each family varied between different slurries and environmental samples. The most abundant OTUs in different slurries affiliated with the genus *Cryptanaerobacter* (76% relative abundance) in the phylum *Firmicutes*, *Desulfobulbus* (59% relative abundance), *Desulfosarcina* (52% relative abundance), *Desulforhopalus* (44% relative abundance) in the phylum *Proteobacteria,* and SEEP-SRB1 clone (45% relative abundance) (Figure 4A). Additionally, the most abundant OTUs (85% relative abundance) in one slurry (SZP6) belonged to the family *Desulfobacteraceae* which could not be assigned to any genus (Figure 4A).

### 3.5. Bacterial Diversity of the Aarhus Bay Original Sediments

Appendix A shows the variation in bacterial communities among sediments originating from the three different biogeochemical zones. All samples were characterized by high bacterial diversity. The sequences retrieved from libraries from the sulfate zone sediment samples affiliated at family level to *Piscirickettsiaceae* (40%), *Shewanellaceae* (8%), *Flavobacteriaceae* (4%), and some other families belonging to the *Gammaproteobacteria*, *Chloroflexi* (9%), and *Alteromonadales* (7%) (Figure 4). The high abundance of *Gammaproteobacteria* (38% of the sequences belonged to *Thiomicrospira*) (Figure 4A) in our study could be related to their ability to oxidize sulfur compounds in the high sulfate and sulfide containing upper sediments. Other culture-independent studies on coastal sediments have consistently detected *Gammaproteobacteria* in high frequencies in upper layers [35,36]. In our study, no sequences related to known SRB were detected in the library from the sulfate zone sediment. Similarly, SRB were not detected by molecular approaches [37] or their abundances were found to be relatively low [5,38] in sulfate zone sediments. On the other hand, SRB were detected by stable isotope probing in pre-enriched sediment slurries [39].

The *Desulfobacteraceae* sequences accounted for 79% of the sequences in the library generated from the SMTZ sediment (Appendix A, Figure 4A). This contrasts with other geochemical zones and indicates that the bacterial community in the SMTZ is distinct from the other zones.

The libraries of 16S rRNA genes from methane zone sediment were dominated by the sequences belonging mainly to *Gammaproteobacteria* (54%) and *Epsilonproteobacteria* (27%) (Appendix A). Previous studies reported that *Gammaproteobacteria* are commonly detected in deep subsurface sediments and the cultivated representatives of this class are often related to the facultative anaerobes [40].

### 3.6. Enrichment of the Propionate Degrading Bacterial Community in Three Geochemical Zones

In marine sediments, propionate can be oxidized by sulfate-reducing bacteria or by consortia of syntrophic bacteria and methanogens in the absence of sulfate [6]. Currently known syntrophic propionate-oxidizing bacteria are affiliated with the class *Deltaproteobacteria* within the phylum of *Proteobacteria* and the class *Clostridia* within the phylum *Firmicutes* [41]. In the enrichments, an apparent shift occurred from the classes *Gammaproteobacteria* and *Epsilonproteobacteria* in sulfate and methane zone sediment samples to the classes *Deltaproteobacteria* and *Clostridia* in SZ and MZ sediment slurries. The relative abundance of *Deltaproteobacteria* increased significantly from 3% to 85% in SZ sediment slurries, and from 0% to 81% in MZ sediment slurries (Appendix A). The high relative abundance of *Desulfobacteraceae* belonging to *Deltaproteobacteria* in SMTZ sediment showed a 2.7-fold increase after enrichment. These results confirm that the members of *Deltaproteobacteria* and *Clostridia* are key in propionate conversion in the presence and absence of sulfate and at various temperatures.

The most abundant propionate-oxidizing sulfate-reducing genera in sulfate-amended slurries were *Desulfosarcina, Desulfofaba* belonging to *Desulfobacteraceae*, and *Desulfobulbus* and *Desulforhopalus* belonging to *Desulfobulbaceae* (Figure 4A). The *Desulfobacteraceae* family mainly consists of sulfate reducers that are complete oxidizers (Rabus et al., 2013) and is commonly found as the dominant SRB in anoxic marine sediments [5,42,43,44]. Surprisingly, unclassified *Desulfobacteraceae* were enriched in the sulfate-free slurry (SZP6), with 85% relative abundance (Figure 4A). Despite *Desulfobacteraceae* being known as a family consisting of sulfate-reducing bacteria, some species are known to be able to have a syntrophic life style. Kendall and coworkers [13] identified an obligate butyrate-utilizing marine syntroph, *Algorimarina butyrica,* belonging to *Desulfobacteraceae*. Similarly, Ulrich and Edwards [45] enriched benzene-degrading cultures which were dominated by a phylotype affiliated to *Desulfobacteraceae*, and were capable of switching from sulfate-reducing to syntrophic life style. These results illustrate the significance of metabolic flexibility of microorganisms under changing conditions such as temperature, electron acceptor availability, the presence/absence of partner organisms.

Sulfate-amended slurries incubated at 10 °C were dominated by *Desulfobulbus* and *Desulforhopalus*, both belonging to *Desulfobulbaceae* (Figure 4A). *Desulfobulbus* had high abundance in both SZ and MZ slurries, whereas the relative abundance of *Desulforhopalus* was rather high in MZ slurries. Most members of *Desulfobulbaceae* are incomplete oxidizers and are specialized in the oxidation of organic acids, including propionate, to acetate [2]. This is in agreement with the observed acetate accumulation in the *Desulfobulbaceae* containing slurries (Figure 1, Figure 2 and Figure 3). *Desulfofaba* belonging to *Desulfobacteraceae*, some of which are psychrophilic marine sulfate reducers capable of propionate oxidation, enriched in SMTZ and MZ slurries incubated at 10°C. Apparently, the *Desulfobulbaceae* and some *Desulfobacteraceae* members had competitive advantage at low temperature, as was supported with statistical analysis result showing a strong negative correlation between the order *Desulfobacterales* and the incubation temperature (*p* < 0.01) (Figure 4B, Appendix A). Enrichment of genera belonging to *Desulfobacteraceae* and *Desulfobulbaceae* in both SZ and MZ slurries indicates the presence of sulfate-reducing bacteria throughout the marine sediment. Previous studies reported the presence of SRB with similar cell numbers in the upper and lower parts of the coastal marine sediments of Limfjorden and Aarhus Bay, Denmark [5,44] and deep-sea sediment [4]. The high abundance of SRB in sulfate-depleted sediments might be due to the acetogenic and fermentative growth characteristics of some SRB in the absence of sulfate [2,12].

*Pelotomaculum* and *Cryptanaerobacter* belonging to the class *Clostridia* dominated sulfate-free and low-sulfate-amended slurries incubated at 25 °C, and were also observed in high sulfate-amended SMTZ slurries (Figure 4A, Appendix A). Currently known propionate-oxidizing syntrophic species of the genus *Pelotomaculum* are *P. schinkii*, *P. thermopropionicum*, and *P. propionicicum* [41]. The *Pelotomaculum* sp. enriched in our study apparently took part in syntrophic propionate conversion. For the *Cryptanaerobacter*, there is to date only one cultured representative, *C. phenolicus*, and it has been observed to utilize phenol and 4-hydroxybenzoate (4-OHB) as an energy source and electron acceptor, and transform them into benzoate [46]. Even though its closest cultured relative is *Pelotomaculum thermopropionicum*, propionate utilization in coculture with a hydrogenotrophic methanogen by this species has not been reported, and other electron donors and carbon sources used by the isolate have not been identified. In a recent study, genomic analysis indicated that *Cryptanaerobacter* sp. G14, which shows 99.9% similarity to *C. phenolicus*, possesses genes for phenol/4-OHB degradation, assimilatory sulfite reduction, and syntrophic propionate degradation [47]. Therefore, we hypothesize that *Cryptanaerobacter* enriched in our study has the capability to convert propionate in syntrophy with hydrogenotrophic methanogens. The statistical analysis showing a positive correlation of the order *Clostridiales* (which *Pelotomaculum* belongs to) with methane and temperature and the strong negative correlation with sulfate and sulfide support our hypothesis (Figure 4B, Appendix A).

*Syntrophobacter* sp. belonging to *Deltaproteobacteria* is capable of either sulfidogenic or syntrophic metabolism during propionate degradation [12]. *Syntrophobacter* species have been identified in shallow marine methanogenic sediments [13,14,15,42]. However, the relative abundance of *Syntrophobacter* sp. in Aarhus Bay sediment was low (<1%). Similarly, very low abundance of *Syntrophobacter*-like sequences at Station 1 in Aarhus Bay were reported [5].

### 3.7. The Archaeal Populations Revealed by Illumina Sequencing

The number of reads per sample for archaeal sequences varied between 5547 and 108810 (Appendix A). The highest percentage of 16S rRNA reads for *Archaea* clustered within the families *Methanomicrobiaceae* (26.6%), an unclassified *Methanomicrobiales* family (EJ-E01) (14.2%), Marine Benthic Group D (MBG-D) and Deep Sea Hydrothermal Vent Eukaryotic Group-1 (DHVEG-1) (13.9%), *Methanosarcinaceae* (7.8%), and unclassified *Methanomicrobiales* (7%) (Appendix A).

### 3.8. Enrichment of the Archaeal Community in Three Geochemical Zones

The relative abundances of the majority of archaeal populations increased after enrichment and variations were observed in terms of both diversity and abundance in each slurry. This indicates establishment of syntrophic associations between syntrophic bacteria and methanogenic *Archaea* after slurries were incubated with propionate. The increase in relative abundance of the order *Methanomicrobiales*, whose members utilize H_2_+CO_2_ and formate as substrates for methanogenesis [48], in almost all slurries suggests that H_2_+CO_2_ and/or formate was the main substrate used by methanogenic *Archaea* in the slurries (Figure 5A, Appendix A). Various genera or groups of *Methanomicrobiales* dominated different slurries based on incubation temperature and sulfate availability. The sequences similar to the unclassified *Methanomicrobiales* clone (EJ-E01) had high relative abundance in three slurries in which the relative abundances of *Methanoculleus* and *Methanogenium* were low. *Methanoculleus* was observed in low sulfate-amended SMTZ and sulfate-free MZ slurries incubated at 25 °C, whereas *Methanogenium* comprised most of the sequences in the libraries of samples from the SZ, SMTZ, and MZ slurries incubated at 10 °C with or without low sulfate amendment (Figure 5A). Given that the optimum growth temperature of *Methanoculleus* ranges between 25 °C and 60 °C, whereas *Methanogenium* species grow best between 15–35 °C, temperature might have been a determining factor. On the other hand, libraries obtained from the high sulfate-amended slurries incubated at 25 °C contained predominantly unclassified *Methanomicrobiales* sequences, termed as EJ-E01 (Figure 5A, Appendix A). The relative abundance of this putative hydrogenotrophic methanogenic group was positively correlated with the presence of sulfate (Figure 5B, Appendix A), suggesting that this group of methanogens could adapt or might have high tolerance to high sulfide concentrations. From the family *Methanosarcinaceae*, *Methanosarcina* was observed to be the dominant genus being present in four methanogenic slurries with read percentages ranging between 14%–30%, whereas *Methanococcoides* had fewer reads in seven slurries (Figure 5A). Despite that *Methanosarcina* species use various compounds, such as acetate, methylated compounds, and H_2_ + CO_2_ [49], they likely constituted close relationships with propionate degraders, and utilized acetate rather than hydrogen. The reason of this might be the competitiveness of hydrogenotrophic methanogenic order *Methanomicrobiales*, explaining their dominance over *Methanosarcina* spp. in slurries. The fact that the families *Methanomicrobiaceae* and *Methanosarcinaceae* were positively correlated to methane and negatively correlated to sulfate (Figure 5B, Appendix A) and their members had high relative abundance in SZP1 and MZP3 slurries where complete propionate conversion occurred (Figure 5A), they played significant roles in hydrogen and acetate consumption, respectively.

*Methanococcoides* (ANME-3 *Archaea*) increased its relative abundance in high sulfate-amended SZ, SMTZ, and MZ slurries (Figure 5A). Lösekann et al. [50] reported that ANME-3 live syntrophically with *Desulfobulbus* spp., which were present in the slurries in which the relative abundance of *Methanococcoides* increased (Figure 4A, Figure 5A). The other anaerobic methanotrophic group that increased its relative abundance was ANME-1 (Figure 5A). This finding indicates that anaerobic oxidation of methane may have occurred not only in the slurries comprised of SMTZ sediment, which was considered as the most common ANME habitat on earth [37,44,51], but also in SZ and MZ sediment slurries.

Marine Benthic Group D (MBG-D) and DHVEG-1 belonging to *Euryarchaeota* had high read percentages in SZ, SMTZ, and MZ sediment samples (Figure 5A). This non-methanogenic archaeal group and Miscellaneous Crenarchaeota Group (MCG) belonging to *Thaumarchaeota* were previously reported to be dominant in deep subsurface sediments in addition to coastal marine surface sediments and that their relative abundances are independent from the major biogeochemical zones, indicating their diverse metabolism [52,53,54,55]. The read percentage of MBG-D and DHVEG-1 increased from 16% in MZ sediment to 45% and 47% in two methane zone slurries containing sulfate (Figure 5A). Webster and colleagues [56] detected MBG-D and MCG in sulfate zone sediment slurry enrichments within the active archaeal community incorporating ^13^C-acetate, resulting in stimulation of MBG-D in acetate-amended slurries. This finding is in line with our observations, where both MCG and MBG-D and DHVEG-1 increased their relative abundances in slurries containing sulfate and their relative abundances were positively correlated with sulfate and acetate concentrations (Figure 5B, Appendix A).

## 4. Conclusions

In this research, we found clear differences in the bacterial and archaeal community in the long-term enrichment slurries depending on the presence/absence of sulfate, incubation temperature, and, in some cases, the biogeochemical zone. Propionate was converted by both sulfate-reducing and syntrophic communities in the presence of sulfate in all depth zones (Figure 6). Members of *Desulfobacteraceae* and *Desulfobulbaceae* were observed in sulfate-amended slurries as well as in sulfate-free slurries at low temperature. This indicates that changing to a syntrophic lifestyle allows sulfate reducers to remain in sulfate-depleted/-limited environments, and explains the high relative abundance of SRB in deep marine sediments. The dominance of *Cryptanaerobacter* at high temperature in sulfate-free slurries inoculated with the SZ, SMTZ, and MZ sediment brought forward the possibility that yet-uncultured species of *Cryptanaerobacter_,_* in addition to *Pelotomaculum* sp., degrade propionate in syntrophy with hydrogenotrophic methanogens. The dominant syntrophic partners belonged to the hydrogenotrophic methanogenic order *Methanomicrobiales*, which enriched in all slurries under different incubation conditions with different representative genera and unclassified groups. *Methanosarcina* enriched only in the absence of sulfate, suggesting a potential competition with acetate-degrading sulfate reducers.

Our results also highlight that several groups of uncultured prokaryotes have an important role in carbon and sulfur cycling in Aarhus Bay sediments. Therefore, future research should focus unravelling the metabolism of these uncultured groups of *Archaea* and *Bacteria*.

## Figures and Tables

**Figure 1 microorganisms-08-00394-f001:**
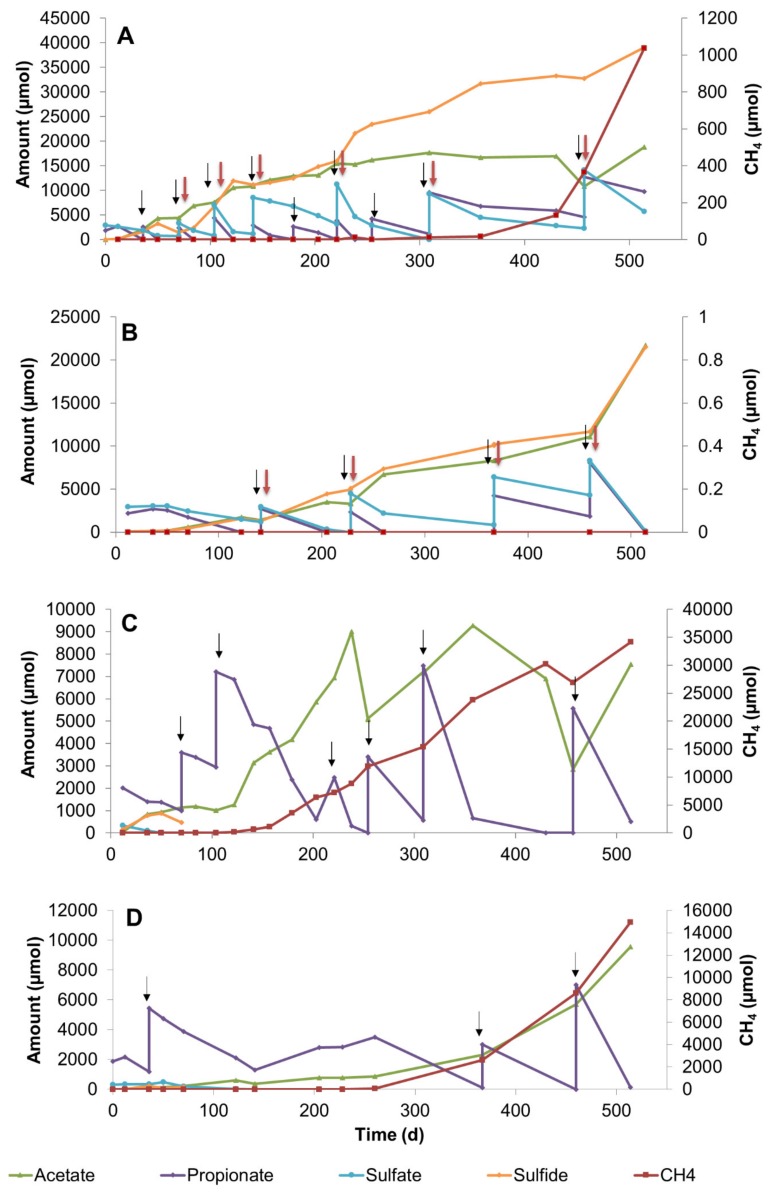
Changes in propionate, sulfate, acetate, sulfide, and methane during 514 days of incubation in sediment slurry enrichments constituted of sulfate zone sediment (**A**) Slurry P4, with 20 mM sulfate addition at 25 °C, (**B**) Slurry P8, with 20 mM sulfate addition at 10 °C, (**C**) Slurry P1, without sulfate addition at 25 °C, (**D**) Slurry P6, without sulfate addition at 10 °C. Arrows denote the time points for the additions of sulfate (red) and propionate (black).

**Figure 2 microorganisms-08-00394-f002:**
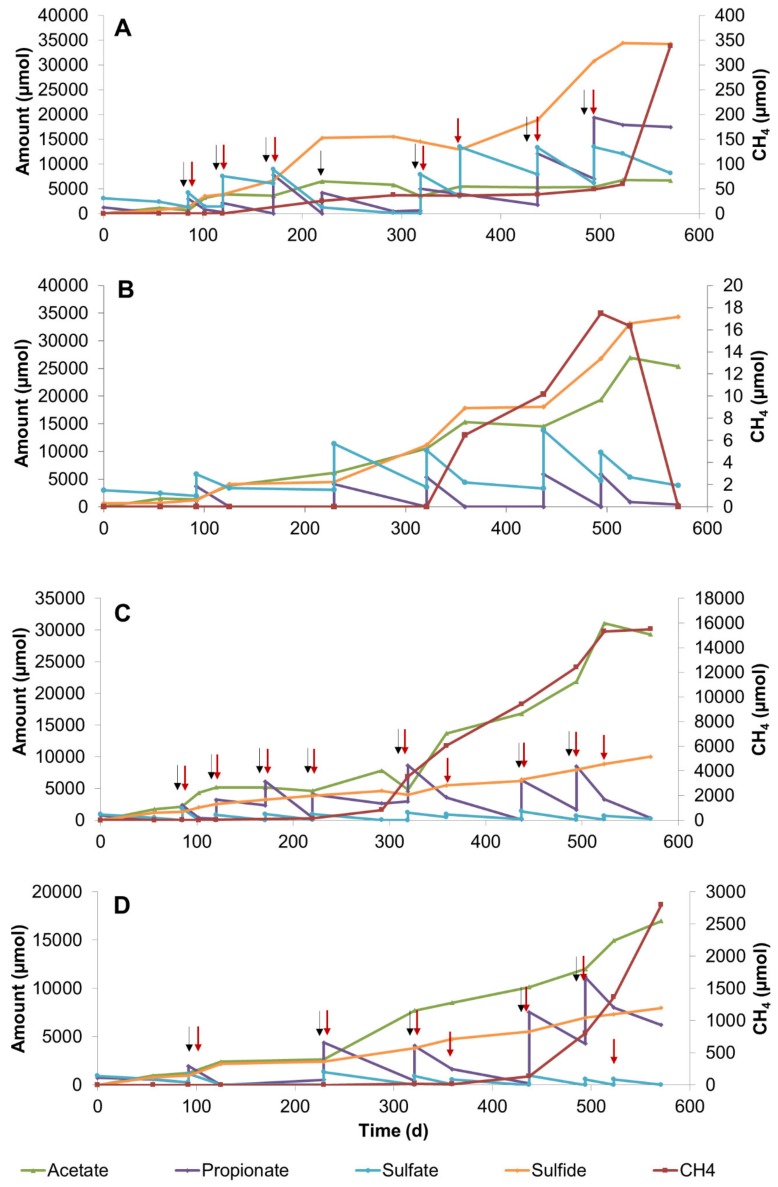
Changes in propionate, sulfate, acetate, sulfide, and methane during 571 days of incubation in sediment slurry enrichments constituted of sulfate–methane transition zone sediment (**A**) Slurry P3, with 20 mM sulfate addition at 25 °C, (**B**) Slurry P7, with 20 mM sulfate addition at 10 °C (**C**) Slurry P2, with 3 mM sulfate addition at 25 °C, (**D**) Slurry P5, with 3 mM sulfate addition at 10 °C. Arrows denote the time points for the additions of sulfate (red) and propionate (black).

**Figure 3 microorganisms-08-00394-f003:**
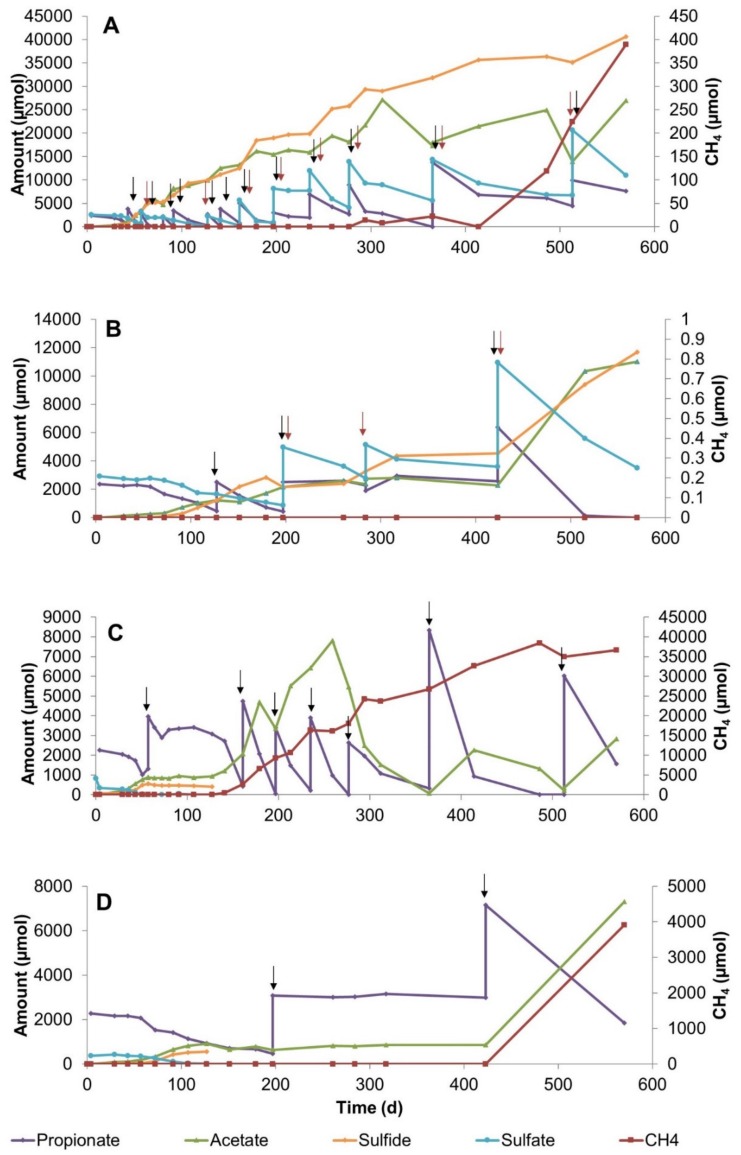
Changes in propionate, sulfate, acetate, sulfide, and methane concentrations during 570 days of incubation in sediment slurry enrichments constituted of methane zone sediment (**A**) Slurry P6, with 20 mM sulfate addition at 25 °C, (**B**) Slurry P8, with 20 mM sulfate addition at 10 °C, (**C**) Slurry P3, without sulfate addition at 25 °C, (**D**) Slurry P1, without sulfate addition at 10 °C. Arrows denote the time points for the additions of sulfate (red) and propionate (black).

**Figure 4 microorganisms-08-00394-f004:**
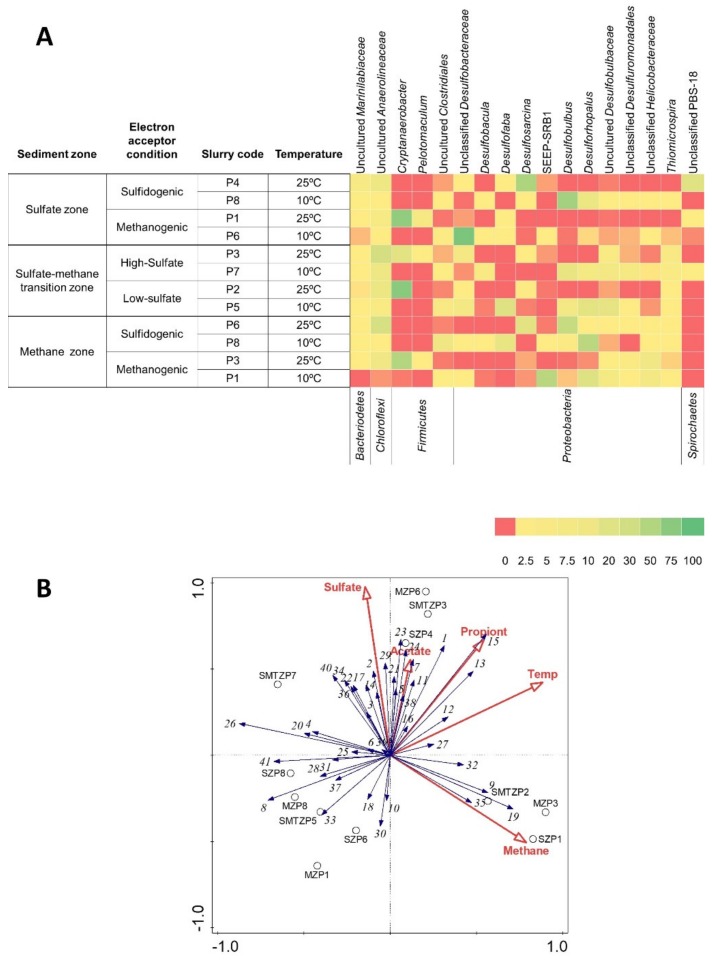
(**A**) The heatmap depicts the relative abundance of the most common (>5%) bacterial 16S rRNA gene sequences across the 12 slurries analyzed and sediment samples belonging to different biogeochemical zones. The heatmap colors represent the relative percentage of the bacterial assignments within each sample. Colors shifted towards dark red indicate higher abundance. Taxonomy is shown at the genus level (unless unclassified) above and at the phylum level below the heatmap. (**B**) Redundancy analysis triplot showing relationship between bacterial community composition at order level and environmental parameters. Environmental variables are given as red vectors. Blue vectors represent bacterial orders. Orders were included with a relative abundance of at least 1% in any sample. Vector length gives the variance that can be explained by a particular environmental parameter. Perpendicular distance reflects association, with smaller distances indicating a larger association. Temp: Temperature, Propiont: Propionate. Operational taxonomic units (OTU) numbers and corresponding taxa are as followed: (1) Bacteria-Other; (2) Actinobacteria-OPB41; (3) Bacteroidetes; (4) Bacteroidetes-BD2-2; (5) Bacteroidetes-Bacteroidales; (6) Bacteroidetes-Flavobacteriales; (7) Bacteroidetes-SB-1; (8) Bacteroidetes-SB-5; (9) Bacteroidetes-Sphingobacteriales; (10) Bacteroidetes-VC2.1.Bac22; (11) Bacteroidetes-vadinHA17; (12) Bacteroidetes-vadinHA17-uncultured bacterium; (13) Candidate division OP9; (14) Chloroflexi-Anaerolineae; (15) Chloroflexi-Anaerolineales; (16) Chloroflexi-GIF9; (17) Cyanobacteria-Chloroplast; (18) Elusimicrobia-Lineage_IV; (19) Firmicutes-Clostridiales; (20) Nitrospirae-Nitrospirales; (21) Planctomycetes-Planctomycetales; (22) Proteobacteria-Rhizobiales; (23) Proteobacteria-Burkholderiales; (24) Proteobacteria-Deltaproteobacteria; (25) Proteobacteria-Desulfarculales; (26) Proteobacteria-Desulfobacterales; (27) Proteobacteria-Desulfovibrionales; (28) Proteobacteria-Desulfuromonadales; (29) Proteobacteria-Sva0485; (30) Proteobacteria-Campylobacterales; (31) Proteobacteria-Gammaproteobacteria; (32) Proteobacteria-Alteromonadales; (33) Proteobacteria-Chromatiales; (34) Proteobacteria-Thiotrichales; (35) RF3; (36) RF3-uncultured bacterium; (37) Spirochaetes-LK-44f; (38) Spirochaetes-PBS-18; (39) Spirochaetes-Spirochaetaceae; (40) TM6; (41) Tenericutes-Acholeplasmatales.

**Figure 5 microorganisms-08-00394-f005:**
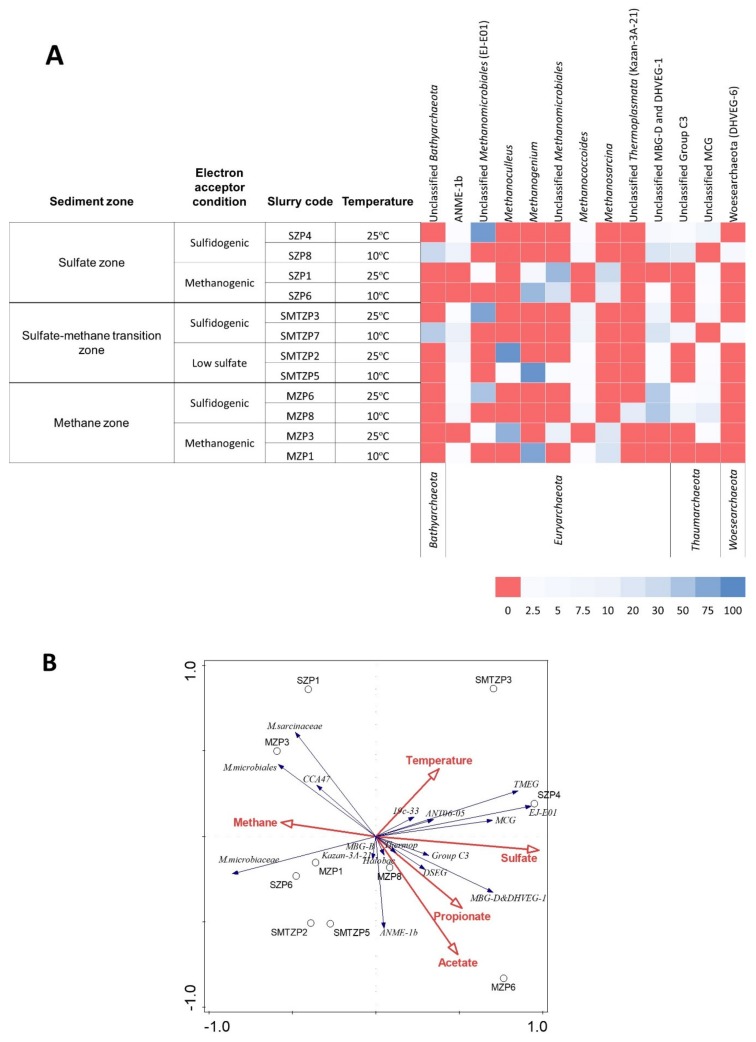
(**A**) The heatmap depicts the relative abundance of the most common (>5 %) archaeal 16S rRNA gene sequences (unless unclassified) across the 12 slurries analyzed and sediment samples belonging to different biogeochemical zones. The heatmap colors represent the relative percentage of the archaeal assignments within each sample. Colors shifted towards dark red indicate higher abundance. (**B**) Redundancy Analysis Triplot showing relationship between archaeal community composition at family level and environmental parameters. Environmental variables are given as red vectors. Blue vectors represent archaeal families. Families were included with a relative abundance of at least 1% in any sample. Vector length gives the variance that can be explained by a particular environmental parameter. Perpendicular distance reflects association, with smaller distances indicating a larger association. Full names of the phylotypes in the plot are as followed: *M.sarcinanaceae*: *Methanosarcinaceae*; *M.microbiales*: *Methanomicrobiales*; *M.microbiaceae*: *Methanomicrobiaceae*; *Halobac: Halobacteriales*; *Thermop*: *Thermoplasmatales*.

**Figure 6 microorganisms-08-00394-f006:**
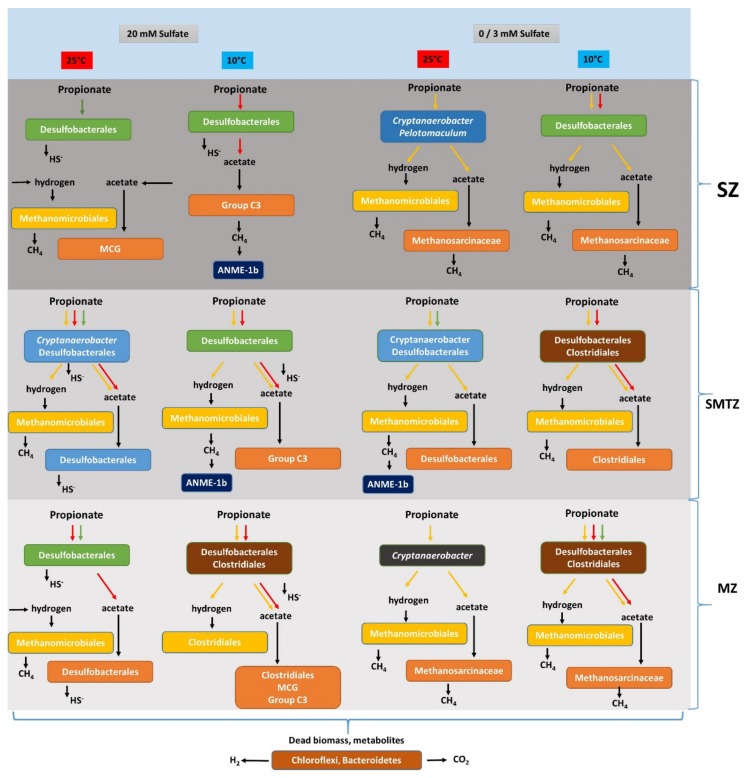
Overview of propionate conversion and the proposed responsible microbial community at different temperatures and sulfate concentrations in enrichment slurries of sulfate (SZ), sulfate–methane transition (SMTZ), and methane zone (MZ) sediment of Aarhus Bay. Putative propionate conversion pathways are shown with different colored arrows; red arrows represent incomplete propionate conversion coupled to sulfate reduction, green arrows represent complete propionate conversion coupled to sulfate reduction, yellow arrows represent syntrophic propionate conversion. Horizontal arrow represents the substrates originate from fermentation, decomposition of dead biomass, and/or metabolites.

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
