# Peer review of "Propionate Converting Anaerobic Microbial Communities Enriched from Distinct Biogeochemical Zones of Aarhus Bay, Denmark under Sulfidogenic and Methanogenic Conditions"

_microorganisms, 2020, doi:10.3390/microorganisms8030394_

Round 1

Reviewer 1 Report

Nicely written manuscript about how an intermediate (propionate) in the carbon degradation pathway is broken down in both sulfate amended and only propionate additions to sediment slurries. The Abstract and Introduction are well written. The former illustrates the manuscript and the Introduction provides a good background. A few minor comments follow.

Materials and methods

Page 2; Line 31, The authors state that the temperature was ~9C. Does this mean the sediment surface temperature? If so this should be clearly stated for the readers. Also is this the same temperature at the bottom of the 3m long gravity cores? This should be presented for the readers.

Just a general question: the authors used 454 sequencing for the bacteria and Illumina sequencing for the archaea. Is there a reason? Why not use Illumina (or 454) for everything?

 Page 3; Line 18, is there a rationale for conducting an additional set of experiments at 25C?

Results and Discussion

Page 12, Line 16, should read '3% to 85%'

Line 17, should read '0% to 81%'

Line 24, shouldn't Rabus et al 2013 be [3]?

Shouldn't everything describing results be in the past tense?

Author Response

Nicely written manuscript about how an intermediate (propionate) in the carbon degradation pathway is broken down in both sulfate amended and only propionate additions to sediment slurries. The Abstract and Introduction are well written. The former illustrates the manuscript and the Introduction provides a good background. A few minor comments follow.

          Thank you for your positive words and suggestions to improve our manuscript.

Materials and methods

Page 2; Line 31, The authors state that the temperature was ~9C. Does this mean the sediment surface temperature? If so this should be clearly stated for the readers. Also is this the same temperature at the bottom of the 3m long gravity cores? This should be presented for the readers.

The temperature is indeed the sediment water interface (SWI) temperature at station M1 of Aarhus Bay, which is typical for May (the month in which the samples were taken). There are no measured data on the sediment core temperature profile. Dale et al., (Seasonal dynamics of the depth and rate of anaerobic oxidation of methane in Aarhus Bay (Denmark) sediments; J Marine Res (2008) 66: 127-155) calculated the T profiles. From this data it can be extrapolated that in the SMTZ the temperature may have been 1 or 2 degrees lower.

Just a general question: the authors used 454 sequencing for the bacteria and Illumina sequencing for the archaea. Is there a reason? Why not use Illumina (or 454) for everything?

We started the project using Pyrosequencing. However, during the project the Pyrosequencing service was phased out and we had to start using Illumina.

Page 3; Line 18, is there a rationale for conducting an additional set of experiments at 25C?

As biological activity is much slower at 10°C, we made a pragmatic decision to include a higher temperature for our slurry incubations to win time.

Results and Discussion

Page 12, Line 16, should read '3% to 85%'

corrected

Line 17, should read '0% to 81%'

corrected

Line 24, shouldn't Rabus et al 2013 be [3]?

corrected

Shouldn't everything describing results be in the past tense?

We have made some changes throughout the document, where appropriate

Reviewer 2 Report

The article by Ozuolmez and co-workers deals with the interactions in propionate-degrading microbial populations obtained from sulfate-containing and sulfate-free sediments of the Aarhus Bay, Denmark. Samples of sediments from three different biogeochemical zones of the Aarhus Bay – from sulfate, sulfate-methane transition and methane zone, were used as inoculum for obtaining enrichments in a medium with propionate amended with sulfate (20 or 3 mM) or without sulfate. Enrichments were incubated at 25 and 10°C for 514–571 days. The products of propionate degradation by enrichments were studied in dynamic. The phylogenetic diversity of Bacteria and Archaea was studied by modern molecular approaches of 16S rRNA gene amplicon pyrosequencing and by Illumina sequencing, respectively, of DNA extracted both from natural samples and from enrichments. Bioinformatics and statistical analysis of the results was carried out, and a generalized scheme was proposed “of propionate conversion and the proposed responsible microbial community at different temperatures and sulfate concentrations in enrichment slurries of sulfate, sulfate-methane transition and methane zone sediment of Aarhus Bay”. The authors obtained abundant data, which may be used for planning further research on the interactions of the microorganisms of the carbon and sulfur cycles in marine environments.

The work was carried out using up-to-date molecular and analytical techniques and may be published in the Microorganisms journal after minor revision.

There are, however, some comments. Propionate is an intermediate product of fermentation of organic substrates. The Aarhus Bay sediments are a well-studied ecosystem. The data on the ranges of propionate concentrations in these sediments or in other marine environments could have been used to support the choice of this substrate.

Since the in situ temperature of sediments was ~9°C, the results obtained by incubation of the enrichments at 10°C may be extrapolated to the natural environment. The authors should provide the reason for the experimental series carried out at 25°C beyond the evident reason that these experiments made it possible to reveal the new and uncultured prokaryotes potentially capable of propionate degradation at elevated temperatures. Simultaneous discussion of all the data set obtained for the enrichments incubated at both 25°C and 10°C make it difficult to understand the results.

Specific comments are listed below.

Global changes: Archaea and Bacteria should be given in italics in the manuscriprt and in Supplementary materials.

  1. 3, lines 4-5: In the formulas, the sign (.) should be replaced by the “·” sign:

Na2HPO4·2H2O, CaCl2·2H2O, MgCl2·6H2O, Na2S·9H2O, Na2HPO4·2H2O, etc.

  1. 6, line 23: The suprescript zero (0) in 100C should be changed to the degree sign (°).
  2. 8, lines 5-6: The phrase: “In low sulfate-amended slurries, on the other hand, the consumed propionate and accumulated acetate concentrations remarkably varied between 25°C (Fig 2C) and 10°C (Fig 2D)” should be changed to: “On the other hand, the consumed propionate and accumulated acetate concentrations in the slurries incubated with low sulfate concentrations at 25°C (Fig 2C) and at 10°C (Fig 2D) differed significantly”.
  3. 10, lines 28-29. The phrase: “In our study, no sequences related to known SRB were detected in the sulfate zone sediment” should be changed to “In our study, no sequences related to known SRB were detected in the library from the sulfate zone sediment”.
  4. 10, lines 33-34. The phrase: “The Desulfobacteraceae that accounted for 79% of the sequences belonged to the SMTZ sediment (Fig S2, Fig 4A)” should be changed to “The Desulfobacteraceae sequences accounted for 79% of the sequences in the library generated from the SMTZ sediment (Fig S2, Fig 4A)”.
  5. 10, lines 36-37. The phrase: “The methane zone sediment was dominated by the sequences belonging mainly to Epsilonproteobacteria (27%) and Gammaproteobacteria (54%) (Fig S2)” should be changed to “The libraries of 16S rRNA genes from methane zone sediment were dominated by the sequences belonging mainly to Gammaproteobacteria (54%) and Epsilonproteobacteria (27%) (Fig S2)”.

A replacement should be made throughout the article: sequences or reads were revealed in the libraries, rather than in enrichment or slurry.

  1. 12, lines 16-17: The phrase: “The relative abundance of Deltaproteobacteria increased significantly from 3% 85% in SZ sediment slurries, and from 0% 81% in MZ sediment slurries (Fig S2)” should be changed to: “The relative abundance of Deltaproteobacteria increased significantly from 3 to 85% in SZ sediment slurries, and from 0 to 81% in MZ sediment slurries (Fig S2)”.
  2. 12, lines 38-39: The phrase: “Most members of Desulfobulbaceae are incomplete oxidizers and specialized in the oxidation of organic acids to acetate, including propionate [2]” should be changed to “Most members of Desulfobulbaceae are incomplete oxidizers and are specialized in the oxidation of organic acids, including propionate, to acetate [2]”.

Figure 1. The quality of panels a–d should be better (similar to that on Fig. 3). The caption should not specify the measuring units (µmol), since it is given on the axes. 20mM should be changed to 20 mM. Figure 1 should be given after its first mention in the text, i.e., on page 6.

Figure 2. The quality of panels a–d should be better (similar to that on Fig. 3). The caption should not specify the measuring units (µmol), since it is given on the axes. 20mM should be changed to 20 mM. On panels a–d, СН4 should be changed to СН4.

Figure 3. The caption should not specify the measuring units (µmol), since it is given on the axes. 20mM should be changed to 20 mM.

Figure 4B. The size of Figure 4B should be as large as possible. In the caption, OTU numbers and corresponding taxa are given as followed: (1)Bacteria-Other; (2)Actinobacteria-OPB41; (3)Bacteroidetes; … The following writing is recommended: (1), Bacteria-Other; (2), Actinobacteria-OPB41; (3), Bacteroidetes;…

Figure 5B. The size of Figure 5B should be as large as possible.

Figure 6. Globally: Methanosarcinacea should be changed to Methanosarcinaceae. The caption should include the explanation of abbreviations: ‘sulfate, sulfate-methane transition and methane zone sediment’ should be changed to ‘sulfate (SZ), sulfate-methane transition (SMTZ), and methane (MZ) zone sediment’.

Figure 6 presents an “Overview of propionate conversion and the proposed responsible microbial community at different temperatures and sulfate concentrations in enrichment slurries of sulfate, sulfate-methane transition and methane zone sediment of Aarhus Bay”.

While the right panel is a relatively detailed presentation of propionate conversion pathways in the absence of sulfate, the left panel looks incomplete, since it does not show the coupling of propionate degradation to sulfate reduction and the final product of sulfate reduction (HS-). The scheme may be supplemented accordingly.

The title of Table S3: “The number of reads per sample generated by Pyrosequencing for Bacteria and HiSeq Illumina sequencing for Archaea” should be specified as: “The number of 16S rRNA gene fragments of Bacteria and Archaea in the libraries generated by Pyrosequencing and HiSeq Illumina sequencing, respectively, for each studied sample".

On Figure S1: The caption should not specify the measuring units (mM), since it is given on the axes. Since on the figure the sampling zones are named in full, explanation of the abbreviations (SZ; Sulfate zone, SMTZ; sulfate-methane transition zone; MZ, methane zone) may be removed. Sulfate & Sulfide (mM) should be changed to Sulfate, Sulfide (mM)

On Figures S2 and S4: The per cent sign (%) is in the axis name and should not be repeated on the axis: 10, 20, 30,... 100 instead of 10%, 20%, 30%… 100%.

In Table S3 and on Figure S2 and Figure S4: It is recommended to specify the "sediment sample belonging to the indicated biogeochemical zone:" Env-CZ, Env-SMTZ, Env-MZ, rather than “Env” for all of them.

On Figure S2: 20mM should be changed to 20 mM. 3mM should be changed to 3 mM.

Information on the sequence data deposition should be given not only on p. 18, but also at the end of the Materials and Methods section, on page 4.

Conclusion section should be transferred to page 14.

  1. 17, line 11: this stud. should be changed to this study.

The authors should follow the accepted guidelines for citation: “Different authors would be separated with semicolons (‘;’). For documents co-authored by a large number of persons (more than 10 authors), you can either cite all authors, or cite the first ten authors, then add a semicolon and add ‘et al.’ at the end”.

The reference 3 should list all authors.

In the reference 6: Eeds. should be changed to Eds.

In the reference 7: Antonie Van Leeuwenhoek should be changed to Antonie van Leeuwenhoek.

In references 55, 56, the period signs in the journal title should be deleted.

In the reference 11: Devol, A.H.. should be changed to Devol, A.H.

In the reference 15: Schink, B. should be changed to Schink, B.;

Rohlin, L.;Gunsalus, should be changed to Rohlin, L.; Gunsalus,

In the references 22, 51 and 53: cite 10 authors and add ‘et al.’ at the end.

In the reference 23: M.;Hugenholtz, should be changed to M.; Hugenholtz,

In the reference 41: Year of publication should be transferred to the end of the reference, before https.

In the reference 53: M.A.;Steen, should be changed to M.A.; Steen,

Stepanaukas,. R.; should be changed to Stepanaukas, R.;

In the reference 54: Cragg, B.A. should be changed to Cragg, B.A.;

Author Response

The article by Ozuolmez and co-workers deals with the interactions in propionate-degrading microbial populations obtained from sulfate-containing and sulfate-free sediments of the Aarhus Bay, Denmark. Samples of sediments from three different biogeochemical zones of the Aarhus Bay – from sulfate, sulfate-methane transition and methane zone, were used as inoculum for obtaining enrichments in a medium with propionate amended with sulfate (20 or 3 mM) or without sulfate. Enrichments were incubated at 25 and 10°C for 514–571 days. The products of propionate degradation by enrichments were studied in dynamic. The phylogenetic diversity of Bacteria and Archaea was studied by modern molecular approaches of 16S rRNA gene amplicon pyrosequencing and by Illumina sequencing, respectively, of DNA extracted both from natural samples and from enrichments. Bioinformatics and statistical analysis of the results was carried out, and a generalized scheme was proposed “of propionate conversion and the proposed responsible microbial community at different temperatures and sulfate concentrations in enrichment slurries of sulfate, sulfate-methane transition and methane zone sediment of Aarhus Bay”. The authors obtained abundant data, which may be used for planning further research on the interactions of the microorganisms of the carbon and sulfur cycles in marine environments.

The work was carried out using up-to-date molecular and analytical techniques and may be published in the Microorganisms journal after minor revision.

Thank you for your positive words on our work and constructive suggestions for improvement.

There are, however, some comments. Propionate is an intermediate product of fermentation of organic substrates. The Aarhus Bay sediments are a well-studied ecosystem. The data on the ranges of propionate concentrations in these sediments or in other marine environments could have been used to support the choice of this substrate.

Thank you for your remark on in-situ concentrations of propionate. These are indeed typically very low. However, the turn-over of propionate may well be higher, but is unknown in in-situ conditions. In our experiments the aim was to unravel the microbial diversity involved in propionate conversion using slurry enrichments. In batch enrichment studies it is quite common to choose for relatively high substrate concentrations. When a chemostat-based slurry enrichment approach would have been used, substrate concertation could indeed have been lower.

Since the in situ temperature of sediments was ~9°C, the results obtained by incubation of the enrichments at 10°C may be extrapolated to the natural environment. The authors should provide the reason for the experimental series carried out at 25°C beyond the evident reason that these experiments made it possible to reveal the new and uncultured prokaryotes potentially capable of propionate degradation at elevated temperatures. Simultaneous discussion of all the data set obtained for the enrichments incubated at both 25°C and 10°C make it difficult to understand the results.

As biological activity is much slower at 10°C, we made a pragmatic decision to include a higher temperature for our slurry incubations to win time.

Specific comments are listed below.

Global changes: Archaea and Bacteria should be given in italics in the manuscriprt and in Supplementary materials.

We presented Archaea and Bacteria in italics, throughout manuscript and supplementary information

  1. 3, lines 4-5: In the formulas, the sign (.) should be replaced by the “·” sign:

Na2HPO4·2H2O, CaCl2·2H2O, MgCl2·6H2O, Na2S·9H2O, Na2HPO4·2H2O, etc.

Corrected

  1. 6, line 23: The suprescript zero (0) in 100C should be changed to the degree sign (°).

Corrected

  1. 8, lines 5-6: The phrase: “In low sulfate-amended slurries, on the other hand, the consumed propionate and accumulated acetate concentrations remarkably varied between 25°C (Fig 2C) and 10°C (Fig 2D)” should be changed to: “On the other hand, the consumed propionate and accumulated acetate concentrations in the slurries incubated with low sulfate concentrations at 25°C (Fig 2C) and at 10°C (Fig 2D) differed significantly”.

Corrected

  1. 10, lines 28-29. The phrase: “In our study, no sequences related to known SRB were detected in the sulfate zone sediment” should be changed to “In our study, no sequences related to known SRB were detected in the library from the sulfate zone sediment”.

Corrected

  1. 10, lines 33-34. The phrase: “The Desulfobacteraceae that accounted for 79% of the sequences belonged to the SMTZ sediment (Fig S2, Fig 4A)” should be changed to “The Desulfobacteraceae sequences accounted for 79% of the sequences in the library generated from the SMTZ sediment (Fig S2, Fig 4A)”.

Corrected

  1. 10, lines 36-37. The phrase: “The methane zone sediment was dominated by the sequences belonging mainly to Epsilonproteobacteria (27%) and Gammaproteobacteria (54%) (Fig S2)” should be changed to “The libraries of 16S rRNA genes from methane zone sediment were dominated by the sequences belonging mainly to Gammaproteobacteria (54%) andEpsilonproteobacteria (27%) (Fig S2)”.

Corrected

A replacement should be made throughout the article: sequences or reads were revealed in the libraries, rather than in enrichment or slurry.

          Thank you for this very constructive remark!

  1. 12, lines 16-17: The phrase: “The relative abundance of Deltaproteobacteria increased significantly from 3% 85% in SZ sediment slurries, and from 0% 81% in MZ sediment slurries (Fig S2)” should be changed to: “The relative abundance of Deltaproteobacteria increased significantly from 3 to 85% in SZ sediment slurries, and from 0 to 81% in MZ sediment slurries (Fig S2)”.

Corrected

  1. 12, lines 38-39: The phrase: “Most members of Desulfobulbaceae are incomplete oxidizers and specialized in the oxidation of organic acids to acetate, including propionate [2]” should be changed to “Most members of Desulfobulbaceae are incomplete oxidizers and are specialized in the oxidation of organic acids, including propionate, to acetate [2]”.

Corrected

Figure 1. The quality of panels a–d should be better (similar to that on Fig. 3). The caption should not specify the measuring units (µmol), since it is given on the axes. 20mM should be changed to 20 mM. Figure 1 should be given after its first mention in the text, i.e., on page 6.

The quality of the panels in the Figure 1 is equal to that of Figure 3 when the .tif files are compared. We assume that in the final version the high quality figures are placed. The positioning of Figure 1 was kept, as it could not fit well in page 6. We leave it to the artistical editors of Microorganisms to decide on the optimal position of Figure 1 in the manuscript.

Figure 2. The quality of panels a–d should be better (similar to that on Fig. 3). The caption should not specify the measuring units (µmol), since it is given on the axes. 20mM should be changed to 20 mM. On panels a–d, СН4 should be changed to СН4.

The quality of the panels in the Figure 2 is equal to that of Figure 3 when the .tif files are compared. We assume that in the final version the high quality figures are placed.; all suggested modifications are made

Figure 3. The caption should not specify the measuring units (µmol), since it is given on the axes. 20mM should be changed to 20 mM.

All suggested modifications are made

Figure 4B. The size of Figure 4B should be as large as possible. In the caption, OTU numbers and corresponding taxa are given as followed: (1)Bacteria-Other; (2)Actinobacteria-OPB41; (3)Bacteroidetes; … The following writing is recommended: (1), Bacteria-Other; (2), Actinobacteria-OPB41; (3), Bacteroidetes;…

We have prepared separate high quality files of Figure 4A and 4B to enable that figure 4B can be presented as large as possible in the final version

We have modified the caption using reviewer’s suggestion

Figure 5B. The size of Figure 5B should be as large as possible.

We have prepared separate high quality files of Figure 5A and 5B to enable that figure 5B can be presented as large as possible in the final version

Figure 6. Globally: Methanosarcinacea should be changed to Methanosarcinaceae. The caption should include the explanation of abbreviations: ‘sulfate, sulfate-methane transition and methane zone sediment’ should be changed to ‘sulfate (SZ), sulfate-methane transition (SMTZ), and methane (MZ) zone sediment’.

          Corrected

Figure 6 presents an “Overview of propionate conversion and the proposed responsible microbial community at different temperatures and sulfate concentrations in enrichment slurries of sulfate, sulfate-methane transition and methane zone sediment of Aarhus Bay”.

While the right panel is a relatively detailed presentation of propionate conversion pathways in the absence of sulfate, the left panel looks incomplete, since it does not show the coupling of propionate degradation to sulfate reduction and the final product of sulfate reduction (HS-). The scheme may be supplemented accordingly.

The scheme in Figure 6 is supplemented according to your suggestions

The title of Table S3: “The number of reads per sample generated by Pyrosequencing for Bacteria and HiSeq Illumina sequencing for Archaea” should be specified as: “The number of 16S rRNA gene fragments of Bacteria and Archaea in the libraries generated by Pyrosequencing and HiSeq Illumina sequencing, respectively, for each studied sample".

Corrected

On Figure S1: The caption should not specify the measuring units (mM), since it is given on the axes. Since on the figure the sampling zones are named in full, explanation of the abbreviations (SZ; Sulfate zone, SMTZ; sulfate-methane transition zone; MZ, methane zone) may be removed. Sulfate & Sulfide (mM) should be changed to Sulfate, Sulfide (mM)

          Corrected

On Figures S2 and S4: The per cent sign (%) is in the axis name and should not be repeated on the axis: 10, 20, 30,... 100 instead of 10%, 20%, 30%… 100%.

Sign has been removed

In Table S3 and on Figure S2 and Figure S4: It is recommended to specify the "sediment sample belonging to the indicated biogeochemical zone:" Env-CZ, Env-SMTZ, Env-MZ, rather than “Env” for all of them.

          In this table the slurry codes refer to the enrichments as presented in Table S2. “ENV” refers to the original sediment sample from the specified zone. We have changed the explanation for ENV  in: “Original Sediment Sample”

On Figure S2: 20mM should be changed to 20 mM. 3mM should be changed to 3 mM.

Corrected

Information on the sequence data deposition should be given not only on p. 18, but also at the end of the Materials and Methods section, on page 4.

Done! Thanks for the suggestion

Conclusion section should be transferred to page 14.

Conclusion has been moved to page 14

  1. 17, line 11: this stud. should be changed to this study.

Corrected

The authors should follow the accepted guidelines for citation: “Different authors would be separated with semicolons (‘;’). For documents co-authored by a large number of persons (more than 10 authors), you can either cite all authors, or cite the first ten authors, then add a semicolon and add ‘et al.’ at the end”.

The reference 3 should list all authors.

“Et al” was removed

In the reference 6: Eeds. should be changed to Eds.

done

In the reference 7: Antonie Van Leeuwenhoek should be changed to Antonie van Leeuwenhoek.

done

In references 55, 56, the period signs in the journal title should be deleted.

done

In the reference 11: Devol, A.H.. should be changed to Devol, A.H.

done

In the reference 15: Schink, B. should be changed to Schink, B.;

Rohlin, L.;Gunsalus, should be changed to Rohlin, L.; Gunsalus,

done

In the references 22, 51 and 53: cite 10 authors and add ‘et al.’ at the end.

done

In the reference 23: M.;Hugenholtz, should be changed to M.; Hugenholtz,

done

In the reference 41: Year of publication should be transferred to the end of the reference, before https.

done

In the reference 53: M.A.;Steen, should be changed to M.A.; Steen,

Stepanaukas,. R.; should be changed to Stepanaukas, R.;

done

In the reference 54: Cragg, B.A. should be changed to Cragg, B.A.;

done

Reviewer 3 Report

this paper gives many informations for the studied sites and permits to better know the relations  between sulfate reducers, syntrophs and methanogens interacting with each other in the conversion of propionate.

interesting and welle written, much work to do that!

Author Response

this paper gives many informations for the studied sites and permits to better know the relations  between sulfate reducers, syntrophs and methanogens interacting with each other in the conversion of propionate.

interesting and welle written, much work to do that!

Thank you very much for your compliments on our manuscript. Indeed, it is representing a lot of work.